# Patient-Related Factors Predicting Stent Thrombosis in Percutaneous Coronary Interventions

**DOI:** 10.3390/jcm12237367

**Published:** 2023-11-28

**Authors:** Larisa Anghel, Bogdan-Sorin Tudurachi, Andreea Tudurachi, Alexandra Zăvoi, Alexandra Clement, Alexandros Roungos, Laura-Cătălina Benchea, Ioana Mădălina Zota, Cristina Prisacariu, Radu Andy Sascău, Cristian Stătescu

**Affiliations:** 1Internal Medicine Department, “Grigore T. Popa” University of Medicine and Pharmacy, 700503 Iași, Romania; larisa.anghel@umfiasi.ro (L.A.); alexandram.clement@gmail.com (A.C.); benchea.laura-catalina@d.umfiasi.ro (L.-C.B.); ioana-madalina.zota@umfiasi.ro (I.M.Z.); cristina.prisacariu@umfiasi.ro (C.P.); radu.sascau@umfiasi.ro (R.A.S.); cristian.statescu@umfiasi.ro (C.S.); 2Cardiology Department, Cardiovascular Diseases Institute “Prof. Dr. George I. M. Georgescu”, 700503 Iași, Romania; leonteandreea32@gmail.com (A.T.); alexroungos@gmail.com (A.R.)

**Keywords:** stent thrombosis, patient-related factors, percutaneous coronary intervention, cardiac death, heart attack

## Abstract

Over the past four decades, percutaneous coronary intervention (PCI) safety and efficacy have significantly improved, particularly with the advent of the drug-eluting stent (DES). First-generation DESs reduced in-stent restenosis rates and targeted lesion revascularization; however, safety issues emerged, due to high incidences of stent thrombosis (ST) linked to death, myocardial infarction, and repeat revascularization. Second-generation DESs were developed to overcome these issues, reducing late-thrombotic-event risk while maintaining anti-restenosis efficacy. Nevertheless, ST still occurs with second-generation DES use. Stent thrombosis etiology is multifaceted, encompassing lesion-, patient-, procedural-, and stent-related factors. Overall, most early-stent-thrombosis cases are linked to procedural and patient-related aspects. Factors like premature discontinuation of dual antiplatelet therapy, resistance to clopidogrel, smoking, diabetes mellitus, malignancy, reduced ejection fraction or undertaking coronary angioplasty for an acute coronary syndrome can increase the risk of stent thrombosis. The aim of this study is to assess patient-related factors that potentially heighten the risk of stent thrombosis, with the objective of pinpointing and addressing modifiable contributors to this risk. By focusing on both patient- and procedure-related factors, a multifaceted approach to coronary revascularization can help minimize complications and maximize long-term benefits in managing ST.

## 1. Introduction

Stent thrombosis (ST), a significant complication during percutaneous coronary intervention (PCI), is defined as a thrombotic occlusion of a coronary stent. ST carries a high risk of morbidity and mortality, typically causing cardiac death or nonfatal heart attacks. While often contrasted with in-stent restenosis, which manifests as chest pain-like symptoms, ST usually presents as an acute condition leading to acute coronary syndrome (ACS) [1]. In 2008, the Academic Research Consortium (ARC) published classifications for ST, divided according to the type of stent, the clinical situation, and the time following stent placement (Table 1) [2].

The etiology of ST is multifactorial, and includes lesion-, patient-, procedural- and stent-related factors. Left main coronary artery or proximal left anterior descending artery lesion, bifurcation or severe calcified lesions and in-stent restenosis are recognized as lesional factors [3]. Procedural factors (incomplete stent apposition, stent underexpansion, residual edge dissection) and stent factors (stent fracture, overlap, long stent length) may also be associated with ST. Overall, most early ST cases are associated with procedural- and patient-related factors. Thus, premature cessation of dual antiplatelet therapy (DAPT), clopidogrel unresponsiveness, smoking, diabetes mellitus, malignancy, lower ejection fraction or coronary angioplasty for an ACS are patient-related factors that may increase the risk of ST [4,5] (Figure 1).

Due to high rates of death, myocardial infarction, and repeat revascularization, ST has been a significant safety concern associated with first-generation drug-eluting stents. To address these issues, second-generation DES with improved stent platforms featuring thinner struts and biocompatible durable or biodegradable polymers were developed. While second-generation DESs have shown better safety outcomes compared to the first-generation ones, cases of ST still occur [6]. Therefore, the present study aims to evaluate the patient-related factors that may increase the risk of ST, with the goal of identifying and correcting modifiable factors that contribute to this risk.

## 2. Methods

ST is a significant clinical occurrence that is linked to elevated death rates. The development of second-generation drug-eluting stents (DESs) aimed to address the challenges associated with lowering the incidence of late thrombotic events, while still ensuring effective prevention of restenosis. However, the occurrence of ST persists, even with the use of second-generation DESs. The etiology of ST is complex, but the earliest ST cases are linked to procedural and patient-related aspects. Due to the limited data available on this topic and the several patient-related characteristics that have been defined, we have chosen to carry out a narrative review, rather than a systematic study. This present study utilizes recent literature published in the past five years to examine patient-related factors associated with an elevated risk of ST. Our objective was to identify and address modifying factors that lead to this risk. A comprehensive study was performed using the existing papers accessible in the MEDLINE, PubMed, and EMBASE databases.

## 3. Premature Cessation of Dual Antiplatelet Therapy

Since the development of PCI, the use of DAPT, which combines aspirin and a P2Y12 inhibitor, has significantly improved the management of ischemic heart disorders. DAPT has significantly decreased the probability of repeated ischemic episodes and, more significantly, coronary ST, although bleeding still poses the biggest threat to the effectiveness of these treatments. The ideal DAPT length following PCI, however, is not known and may range among patient groups. Clinical research involves determining the best duration for a given risk-to-benefit ratio. Whether treated invasively or conservatively, the European Society of Cardiology advises that DAPT be maintained for 12 months in patients with ACS [1]. Nevertheless, P2Y12 inhibitors can be stopped after six months in individuals with a significant risk of bleeding. The suggested DAPT duration with the use of modern drug-eluting stents is six months after PCI in patients with stable ischemic heart disease and only three months in patients with bleeding concerns. A significant risk factor for ST is the premature termination of antiplatelet treatment, and this connection depends significantly on both the circumstances surrounding DAPT cessation and the time frame following PCI. Three types of DAPT cessation have been described in the literature: discontinuance, interruption, and disruption. Physician-recommended discontinuation of antiplatelet therapy in individuals deemed to require DAPT is no longer referred to as discontinuation. Interruption was defined as the temporary suspension of anti-platelet treatment owing to a medical emergency, followed by the immediate reinstitution of DAPT within 14 days. DAPT disruptions included DAPT termination because of bleeding or disobedience [1,3,7].

Most studies comparing shorter DAPT durations (3–6 months) and normal DAPTs (between 6 and 12 months) lacked the necessary power to identify variations in ST rates. As a result, care should be used when interpreting the clinical results of these randomized controlled trials involving ST. Additionally, most of these trials only included a limited percentage of patients with ACS and low-risk individuals. A meta-analysis, which particularly addressed this issue, found that 3 months of DAPT after ACS, but not after PCI for stable coronary artery disease (CAD), was linked to a greater incidence of ST and myocardial infraction. In a subanalysis of The Dutch ST Registry, where 74% of patients had ACS as the reason for index-PCI, the overall rate of ST after stopping clopidogrel was 4.6% (95%CI: 3.9–5.4%), as opposed to 1.7% (95%CI: 1.5–1.9%) in patients who continued taking the medication. Additionally, the occurrence rate of ST was found to be 35.4% when the use of clopidogrel was ceased during the initial 30 days following the index PCI and 11.7% when clopidogrel was discontinued within the first 180 days [8,9].

A retrospective observational cohort study that included individuals with ACS between January 2008 and December 2013, and who received DAPT followed by antiplatelet monotherapy, evaluated the rate of cardiovascular events after DAPT cessation and investigated indicators of events during long-term follow-up after these processes. It established that discontinuation of DAPT is linked to an increased risk of cardiovascular disease. The incidence of cardiovascular events was not greater in the ACS cohort in the early post-DAPT cessation phase compared to later periods, indicating no rebound increase in risk in this population. In the long-term follow-up period following the completion of DAPT treatment, cardiovascular events took place. It has been discovered that age, the absence of revascularization therapy for the ACS, and the length of DAPT treatment are all independent predictors of cardiovascular events [10]. On the other hand, studies that examined patients who received short-duration DAPT that included aspirin and a P2Y12 inhibitor followed by only a P2Y12 inhibitor did not find any differences between short- and standard-duration DAPT regarding efficacy outcomes (all-cause death, major adverse cardiovascular events (MACE), myocardial infarction, stroke, and ST), but did find that major bleeding events were significantly reduced with short-duration DAPT. Therefore, following PCI, short-term DAPT followed by P2Y12 inhibitor monotherapy is a viable alternative that can be used, especially in patients who have an elevated risk of bleeding [3].

One month of DAPT was shown to be non-inferior to continuing therapy for a minimum of two more months within the MASTER DAPT (The Management of High Bleeding Risk Patients Post Bioresorbable Polymer Coated Stent Implantation with an Abbreviated Versus Prolonged DAPT Regimen), with respect to the incidence of net adverse clinical events and significant adverse cardiac or cerebral events. The findings of the GLOBAL LEADERS trial indicate that the use of a one-month DAPT followed by 23 months of ticagrelor alone does not result in a reduced risk of new Q-wave myocardial infarction or a lower occurrence of all-cause mortality when compared to a 12-month DAPT followed by 12 months of aspirin monotherapy. The STOPDAPT-2 (Short and Optimal Duration of Dual Antiplatelet Therapy After Everolimus-Eluting Cobalt-Chromium Stent-2) trial found that in PCI patients, one month of DAPT followed by clopidogrel monotherapy had a much lower rate of cardiovascular and bleeding events than 12 months of DAPT with aspirin and clopidogrel. Additionally, a recent meta-analysis of these studies found no significant differences between the rates of all-cause mortality, stroke, MI, and ST during the follow-up period of 1 month to 1 year when comparing one-month DAPT to routine DAPT in patients undergoing PCI [4,11,12,13].

In a subanalysis of the PARIS study, patients at high atherothrombotic risk (HATR), defined as those with a history of myocardial infarction, a history of stroke, or peripheral artery disease (PAD), were compared to patients at low atherothrombotic risk (LATR), who were defined as those without a history of any of these conditions. Regardless of their clinical presentation, HATR patients who have a coronary DES implantation are more likely to experience later cardiovascular events. Because of the significant risk of cardiovascular events in these patients, doctors were more inclined to maintain DAPT in these individuals. However, there was no increased incidence of adverse outcomes among individuals with HATR who stopped taking DAPT as advised by their doctor after 2 years post-PCI. In patients with or without HATR, interruption of DAPT owing to patient noncompliance or bleeding was linked to an increased risk of cardiovascular events. This finding gives doctors performing the treatment comfort when they must decide whether to stop administering DAPT because of other concurrent clinical considerations, and it supports the idea that bleeding prevention measures should be implemented, regardless of the atherothrombotic risk [14]. Another subanalysis of the same PARIS trial evaluated the incidence and relationships between modalities of DAPT discontinuation and outcomes across bleeding risk categories (high, moderate, and low), which were classified according to their risk of bleeding, using the PARIS bleeding risk score. Patients with high bleeding risk (HBR) were older, more likely to have concomitant conditions, and experienced more ischemia and bleeding episodes than non-HBR patients. Additionally, DAPT interruption was linked to an elevated risk of severe adverse cardiac events across all bleeding risk categories, but physician-guided DAPT termination was not. This was true for both HBR and non-HBR patients. For either method of cessation, there was no relationship between bleeding risk status and clinical outcomes [15].

Furthermore, even if patients with chronic kidney disease (CKD) exhibit higher risks of death, ischemic, and bleeding complications at 2 years after PCI compared with their non-CKD counterparts, and physicians are more likely to interrupt DAPT in the first year after PCI among this category of patients, risks after DAPT cessation (irrespective of underlying mode) are not modified by the presence or absence of CKD [16].

Older individuals, another specific group of patients who receive coronary procedures, are more likely to experience ischemic events and are less likely to tolerate extended DAPT, due to the risk of bleeding. In patients >75 versus 75 years old from the MASCOT registry, Chandrasekhar et al. evaluated 1-year clinical outcomes and DAPT cessation events, and they found that while TLF (target lesion failure, composite of cardiac death, myocardial infarction) incidence was similar in both groups, bleeding rates and physician-guided DAPT discontinuation rates were significantly higher in elderly patients. Regardless of age, physician-driven DAPT withdrawal was not linked to a poorer 1-year TLF following COMBO stenting [17]. Similarly, in the two years following PCI, elderly patients’ rates of DAPT discontinuation were greater than those of younger age groups. The effects of each DAPT cessation method varied greatly, according to age. Disruption was linked to an elevated cardiovascular risk in younger patients but not in individuals over the age of 75. Discontinuation and interruption were not linked to an increased cardiovascular risk across age groups [18].

The benefit of continuing DAPT beyond 6 to 12 months following PCI with second-generation drug-eluting stents was demonstrated by a cohort study. The rates of all-cause and cardiac mortality were reduced in people who stopped DAPT after 9 months following PCI, compared to subjects who maintained the drug, but this was not the case for vascular or noncardiovascular death. This decrease in cardiac mortality rates without any correlation to noncardiac causes of death shows that the effect of stopping DAPT after this period is limited to cardiac events, and is generally safe. A lower incidence of severe bleeding and MI were also linked to stopping DAPT after 9 months. On the other hand, when DAPT was stopped earlier than 9 months, the risk of cardiovascular and noncardiovascular mortality was greater (HR, 2.03–3.41), which is probably due to variables that cause early death [19]. According to Silva et al., the most significant factor associated with ST was the early discontinuation of clopidogrel (used for less than 12 months in cases of MI, regardless of the type of stent, less than 30 days in patients undergoing elective PCI using BMS, and less than 12 months using DES). Hypertension, dyslipidemia, smoking, obesity, previous myocardial infarctions, bifurcation lesions, and multiple stents were also associated with ST [20].

Even though increased platelet reactivity, aggregation, and hypercoagulability in DM patients lead to the pro-thrombotic state, raising the risk of ACS and a recurrence of thrombotic events, a study that included ACS diabetic patients enrolled in the REDUCE trial treated with the COMBO stent, and randomly allocated to either 3 or 12 months of DAPT, discovered that, at any given point during follow-up, outcomes occurred at similar rates. Similar outcomes were discovered for particular outcome measures, such as ST, hemorrhage, recurrent MI, and death [21].

## 4. Percutaneous Coronary Intervention for Acute Coronary Syndrome

ST-segment elevation myocardial infarction (STEMI) patients with ACS should have primary PCI, under current recommendations, to minimize mortality and morbidity. ST is currently the most prevalent post-procedural complication of PCI, owing to coronary artery stents, which have poor clinical results and high in-hospital and long-term mortality rates. Patients suspected of ST should undergo coronary angiography to confirm the diagnosis, with primary PCI being the preferred treatment, as most ST patients have STEMI or non-STEMI [5,22,23,24]. The risk of ST was found to be significantly higher in people with STEMI compared to people with stable CAD, and a number of lesion-related or patient-related factors (vessel size, lesion length, ACS or unstable angina, left anterior descending artery (LAD) involvement, presence of a thrombus, plaque characteristics, coronary blood flow, advanced age, local platelet/coagulation activity, left ventricular ejection fraction (LVEF), PAD, renal failure, diabetes mellitus (DM), and procedural and post-procedural factors (stent malposition, stents under expansion, undersized stents, residual dissection), including lesion characteristics, stent type, and thrombus burden, were associated with the elevated risk of ST [6,25,26,27,28].

Insufficient platelet inhibition, which results from clopidogrel hyporesponsiveness, noncompliance with antiplatelet treatment (APT), or suspension of APT for unscheduled or non-deferrable surgery, is another important risk of ST. There are studies that showed a temporal relationship between prematurely stopping clopidogrel and ST within the first 6 to 9 months after DES implantation and no correlation thereafter, while others, such as the PARIS registry, showed that 74% of ischemic events, including ST, happened while patients were on DAPT [6,29].

In a large meta-analysis involving STEMI patients, DES outperformed BMS in terms of long-term effectiveness. There were no differences between certain DES types in terms of long-term effectiveness and safety. In lowering ST, second-generation DESs exceeded first-generation DESs [30]. However, a retrospective, non-randomized observational study that followed 1460 consecutive PCI patients found that at five years, rates of MACE were significantly lower in the everolimus-eluting stent (EES) group, compared to the sirolimus-eluting stent (SES) group [31]. A retrospective analysis of 569 STEMI patients found that ST is more common in males, hypertensive patients, diabetics, and those with poorer pre-PCI LVEF and a classification of Killip Class. ST patients had a 36.4% in-hospital death rate, compared to 0.2% for those without ST. Acute or sub-acute ST occurred in 5.8% of patients. In the same way, a prospective observational study has shown that focusing on male gender and pre-procedure thrombolysis in myocardial infarction (TIMI) flow 0 may identify and treat highly sensitive people [23,32]. Stent thrombosis risk score (STRS) is an independent predictor of ST after primary PCI, and has statistically significant discriminating power, along with Killip Classes III-IV at presentation, according to Kumar et al., who sought to evaluate the reliability of STRS in predicting early ST. As a result, early ST after primary PCI can be risk-stratified using STRS [33].

In 2303, in patients receiving primary PCI, the incidence of early ST—defined as ST occurring within 30 days after coronary stent deployment—was 1%, with increased in-hospital and 30-day mortality [34]. Additionally, a meta-analysis of 23 trials evaluated the short- and long-term clinical outcomes of PCI for early stent thrombosis (EST) vs. late stent thrombosis (LST) and very late stent thrombosis (VLST). In-hospital, 30-day, 1-year, and long-term death rates were considerably higher in the EST group, as compared to mortality rates in the late and very-late thrombosis (LST/VLST) groups. Therefore, it was determined that after PCI therapy, patients with EST had worse clinical outcomes than those with LST/VLST in both short- and long-term follow-up [35].

## 5. Smoking

The short-term prognosis for smokers vs. nonsmokers using clopidogrel following PCI is still unknown, despite the knowledge that smoking is a significant contributing factor to recurrent MI and mortality after PCI [36]. Cohort research that used data from the Korea AMI Registry (KAMIRT) demonstrated that the cumulative incidence of all-cause death, both early (30 days) and late (1 year and 2 years), as well as the cumulative incidence of Re-MI after 1 month of the index PCI, were higher in the female group than the male group after adjustment, strongly indicating a gender difference for the major clinical outcomes [37,38]. Additionally, there was no significant difference in outcomes among smokers who had STEMI and those who had NSTEMI [39,40].

Previous research has discussed the “smoker’s paradox”, whereby smokers experience lower rates of adverse events following MI and after PCI, due to pharmacokinetic and pharmacodynamic interactions between current smoking and clopidogrel metabolism, which result in higher levels of the active clopidogrel metabolite, as well as a stronger platelet inhibition among current tobacco users when compared to nonsmokers. The differing features between nonsmokers and current smokers who report symptomatic CAD may help to explain the reduced incidence of adverse events seen among current smokers in earlier research. Current smokers tend to be younger and have fewer risk factors than nonsmokers at the time of CAD manifestations. A logistic regression model found that the OR of high platelet reactivity (HPR) is 4.9 times greater in the smoking cessation group than in the control group, in patients treated with 75 mg/day of clopidogrel after successful PCI, who reported smoking more than 10 cigarettes per day. This lends considerable credence to the hypothesis that HPR frequency increases after quitting smoking. Furthermore, this impact is unrelated to the initial P2Y12 reaction units (PRU). Conversely, it was discovered that smokers do have a lower P2Y12 reaction unit than non-smokers, in an analysis of the ADAPT-DES study (Assessment of Dual Antiplatelet Therapy with Drug-Eluting Stents), the largest study assessing platelet reactivity after percutaneous coronary intervention, and analysis of its relation to clinical outcomes. This finding confirms an enhanced antiplatelet effect, on average, among smokers. Additionally, two years after percutaneous coronary intervention, smoking was an independent predictor of all-cause death. Thus, the smoker’s paradox is not supported by these studies. Also, it was found that smokers with high platelet reactivity during therapy had poor results. For instance, smokers with significant on-treatment platelet reactivity had a greater likelihood of ST [36,41].

Smokers initially looked to have better clinical outcomes after an AMI, but these positive benefits quickly vanished, and, after adjustment, the all-cause mortality was greater in the smoking group. Additionally, clinical outcomes (a composite of all-cause mortality, non-fatal MI, any revascularization, cerebrovascular accident, rehospitalization, and ST) among the smoking group tended to decline as smoking intensity rose. These findings demonstrate the falsity of the “smoker’s paradox”, and validate the necessity of assisting patients with AMI in quitting smoking [42].

A retrospective study carried out on 50 patients between the ages of 18 and 50 who were admitted with ACS and underwent re-catheterization within a year of their initial cardiac catheterization revealed that, among those who went through a new infarction, 14% had ST, while 12% had stent restenosis, and among cardiovascular risk factors, dyslipidemia, hypertension, and smoking were associated with a high risk of undergoing re-catheterization [43]. Following PCI with stenting, extended DAPT significantly reduced the incidence of MI and probable or certain ST among smokers and nonsmokers, according to a meta-analysis that included randomized controlled trials (RCTs) of extended DAPT (>12 months) compared with DAPT for 6–12 months. Additionally, distinct effects on MACCE (substantially decreased among smokers) and bleeding (increased among non-smokers) with prolonged DAPT were seen [44]. Smoking status and a two-stent strategy significantly increased the risk of ST, nearly tripling it, according to a retrospective study that examined the BIFURCAT (comBined Insights From the Unified RAIN and COBIS bifurcAtion regisTries) registry on coronary bifurcations [45].

The smoker’s paradox following PCI with coronary stent placement is also substantially discounted by the findings of a large-scale investigation. Active smokers had an increased incidence of MI and ST following stenting over the course of five years, even before multivariable correction. Smoking was also connected to higher long-term rates of cardiac and all-cause death when baseline imbalances were also taken into consideration. Smoking is a significant predictor of worse outcomes following PCI, according to this large, patient-level, pooled analysis with a 5-year follow-up [46].

## 6. Lower Ejection Fraction

Left ventricular (LV) systolic dysfunction is associated with a gradual rise in major adverse cardiovascular events (MACEs). PCI performed on patients with reduced LVEF leads to higher short-term and long-term mortality rates, an increased occurrence of nonfatal myocardial infarction and a greater requirement for repeat revascularization procedures. The presence of more intricate clinical and angiographic characteristics in this specific patient group may contribute to the higher occurrence of unfavorable outcomes [47].

An LVEF of less than 30%, along with congestive heart failure, are variables included in the DAPT score. This score emerged as one of the initial clinical risk-assessment tools to effectively distinguish the linked risks of bleeding and ischemic events in patients undergoing PCI [48].

The occurrence of ST demonstrates a notable rise as LV function declines, with this association being observed across different types of ST (definite and probable), as well as in both early and late cases. However, this association is primarily seen in patients who have at least moderate systolic dysfunction, as indicated by an estimated LV ejection fraction (LVEF) of ≤40% [47,49,50]. Lower LVEF is correlated with both EST and LST. In particular, EF of less than 30% significantly increases the risk of early ST [51].

The precise mechanism behind the elevated rates of ST in patients with systolic dysfunction remains uncertain. A hypothesis suggests that reduced shear forces within the stented segment occur as LV function decreases. This altered hemodynamic environment may contribute to a higher propensity for the development of ST [52].

In a study that included 5377 patients undergoing PCI, an LVEF ≤ 40% was associated with a higher risk of ST. Patients with lower LVEF also experienced a higher incidence of in-hospital bleeding and vascular complications, which may have led to the discontinuation of antiplatelet medications. Additionally, the prevalence of chronic renal insufficiency, a recognized predictor of ST, increased as LVEF deteriorated [47].

Another possible explanation is the association between low LVEF and high on-treatment platelet reactivity (HPR). Studies have demonstrated that HPR serves as a risk factor for subsequent cardiovascular events in patients undergoing treatment with aspirin and/or clopidogrel. Elevated platelet reactivity, despite medication use, indicates a reduced effectiveness of antiplatelet therapy, potentially leading to an increased likelihood of experiencing secondary cardiovascular events [53].

The antiplatelet effects of clopidogrel in vivo depend on the concentration of its active metabolite in the bloodstream. Therefore, any factors that influence drug absorption and metabolic activation, such as CYP450 activity, can impact the effectiveness of the medication [54]. Heart failure can affect both processes. Autonomic disturbances associated with heart failure, such as increased sympathetic activity and decreased parasympathetic activity, as well as tissue hypoperfusion, can lead to reduced gastrointestinal motility and slower transit time. Consequently, oral absorption may be delayed, resulting in lower peak plasma concentrations of the drug [55]. Furthermore, heart failure is characterized by a decrease in total hepatic blood flow, proportionate to the reduction in cardiac output. Liver congestion and hypoperfusion can occur in heart failure, potentially impairing hepatic drug metabolism, including the metabolism of clopidogrel [56]. Additionally, chronic hypoxemia, a common feature of heart failure, can modulate the activity of CYP450 enzymes, which further adds to the complexity of drug metabolism in these patients [57]. Overall, these factors associated with heart failure, such as autonomic disturbances, tissue hypoperfusion, liver congestion, and chronic hypoxemia, can impact the absorption, metabolism, and effectiveness of clopidogrel, leading to a higher risk of ST [58].

Interestingly, the use of drug-eluting stents does not appear to impact the risk of ST. Furthermore, based on the available data, there is no specific degree of systolic dysfunction that suggests the preference of one stent type over the other, for implantation [47].

When it comes to ST in the context of acute vs. chronic coronary syndrome (CCS), patients with ACS tend to have a lower LVEF [22,59]. Reduced LVEF, along with age and creatinine, is also part of the ACEF score, which predicts poor outcomes after ACS, including ST. This score serves as a straightforward and reliable tool for risk stratification and proves its validity when applied to a diverse patient population with ACS who are referred for coronary angiography [60]. Low LVEF also appears to be an independent risk factor for mortality in these patients, along with some other variables such as advanced age and multivessel disease. However, based on the current evidence, it is still uncertain whether ST leads to greater ischemic myocardial damage in patients with CCS, compared to ACS [61].

## 7. Malignancy

Cancer is recognized as an acquired thrombophilic condition, whereby individuals affected by it experience an increased propensity for thrombosis. Hence, the occurrence of an increased incidence of thrombotic events, particularly ischemic stroke, in cancer patients should not be unexpected. Notably, cancer patients commonly exhibit a hypercoagulable state, characterized by a heightened predisposition to blood clot formation, even in the absence of clinically evident thrombosis. Furthermore, there is emerging evidence suggesting that clotting activation may play a contributory role in tumor progression [62].

The pathogenesis of thrombosis in cancer is multifactorial in nature, involving a complex interplay of various mechanisms. Notably, tumor cells possess the capacity to interact with and activate the host hemostatic system, thereby contributing to the development of thrombotic events. This interaction between tumor cells and the hemostatic system has been assigned as a relevant factor in the pathogenesis of cancer-associated thrombosis [63].

The pro-thrombotic nature of several chemotherapeutic agents is well recognized, and, historically, the documentation of the association between chemotherapy and arterial thrombosis has largely relied on case reports. While it remains challenging to precisely determine the individual contribution of chemotherapy-induced prothrombotic effects versus the underlying hypercoagulable state of malignancy, numerous chemotherapeutic agents have been linked to a notable incidence of arterial thromboembolic events. Among the most-frequently observed arterial thrombotic events are myocardial infarction, cerebrovascular events, and peripheral artery disease [64,65].

It is theoretically plausible that cytostatic and cytotoxic chemotherapeutic agents may hinder the process of stent endothelialization. This impairment could have implications for the growth of both the surrounding endothelial cells and the circulating progenitor cells, which play a crucial role in stent endothelialization. Notably, patients who experience ST demonstrate a diminished capacity for forming endothelial progenitor colonies [66]. In individuals with cancer, the levels of endothelial and circulating progenitor cells are often suppressed, particularly during the acute phase of treatment, and may remain lower in those receiving therapies targeting the vascular endothelial growth factor (VEGF) [67,68].

Besides chemotherapy, cancer patients are also treated with various radiotherapy regimes. The incidence of ischemic heart disease is elevated in breast cancer patients who undergo radiotherapy, specifically due to the exposure of the heart to ionizing radiation. This elevated risk is directly related to the average radiation dose received by the heart, and becomes evident within a few years following exposure, persisting for a minimum of 20 years. Women with pre-existing cardiac risk factors experience more substantial absolute increases in risk resulting from radiotherapy, compared to women without such risk factors [69]. External beam radiation therapy (EBRT) is a fundamental component of cancer treatment. However, when utilized for specific thoracic malignancies such as breast, lung, Hodgkin and non-Hodgkin lymphoma, and esophageal cancer, it results in considerable cardiac exposure. However, the administration of thoracic EBRT does not appear to elevate the risk of clinically significant stent failure in cancer patients who have undergone a coronary artery stenting either prior to or after EBRT. Moreover, PCI with stents can be safely employed as a treatment modality for CAD in cancer survivors who have previously received EBRT [70].

The occurrence of thrombotic events in cancer patients bears significant clinical implications, influencing the overall morbidity and mortality associated with the underlying malignancy. This condition encompasses patient frailty, a pro-thrombotic phenotype, and a heightened risk of bleeding that can disrupt DAPT protocols [71,72].

Malignancy is therefore considered an independent, patient-related risk factor for both EST and LST. It is especially associated with VLST. Moreover, a history of malignancy emerges as the most significant clinical factor associated with LST [51,73].

In a retrospective study, the incidence of ST in patients with established malignancies who received bare-metal stents (BMSs) was found to be 5.56%. The median time from stent implantation to the occurrence of ST was reported to be 7 days. It is noteworthy that most of these patients (83.3%) were receiving DAPT at the time when ST was observed [72].

Cancer patients who undergo PCI have a higher rate of myocardial infarction (MI) and ST, compared to non-cancer patients. The increased risk of MI persists over the 5-year follow-up period, with sudden cardiac death and ST being the most common types of MI in cancer patients. The rate of ST in cancer patients is nearly three times higher, and the risk is present for both BMS and drug-eluting stents. The elevated risk of ST is primarily observed within the first year after PCI. Additionally, cancer patients have a higher rate of repeat revascularization, indicating the need for additional procedures. Metastatic cancer and a high DAPT score are independent predictors of future non-type 2 MIs among cancer patients, and high-thrombotic-risk cancers are associated with a greater risk of ST in the first year after PCI [74].

## 8. Diabetes Mellitus

The presence of diabetes is a well-established risk factor for coronary artery disease, and its role as an adverse-event contributor after percutaneous coronary intervention with DES, as well as bare-metal stents (BMSs), has been extensively studied [75]. In this new era of reperfusion therapy, significant progress in stent technology and medical therapy has improved outcomes; however, an increased risk of MACE and cardiovascular mortality is still present in this population of patients. Several factors are associated with this increased risk of ischemic events, such as increased platelet aggregation, a pro-thrombotic state due to platelet hyperactivity, and endothelial dysfunction. ST is a serious complication of DES implantation, regardless of timing (early, late, or very late).

Several pathophysiological mechanisms affecting short- and long-term outcomes in patients with diabetes have been discovered. Hyperglycemia and insulin resistance play an important role in accelerating atherosclerosis, promoting endothelial dysfunction, impairing vasodilation, and exaggerating neointimal hyperplasia [76]. Furthermore, diabetic patients frequently exhibit a proinflammatory status, which in turn enhances vascular proliferation as a response to arterial injury mediated by stent placement [77].

The complete mechanisms for increased rates of ST in diabetic patients are yet to be fully elucidated, but newer data point towards platelet hyperreactivity playing a significant role. In patients with diabetes, three independent but frequently interrelated variables, high blood glucose, oxidative stress, and elevated vascular shear forces, coexist and create a complicated association of risk factors. The convergence of these factors leads to an increased level of pretreatment platelet activity, and consequently a worse response to P2Y12 inhibition, which may increase the risk of ischemic events [78]. In the diabetic population, randomized data have shown that, besides the hyperactivity of platelets, the platelets themselves may be less responsive to antiplatelet therapy. In both in vitro and in vivo studies, platelet reactivity testing after clopidogrel administration has been shown to be less downregulated in diabetic, compared to nondiabetic, patients [79].

The common anatomical patterns of CAD in diabetic patients may also play an important role in their prognosis and response to revascularization. Angiographic and autopsy studies have shown that patients with diabetes more frequently present with left main coronary artery lesions, multivessel disease, and diffuse vascular involvement [80]. Diabetic patients often present narrower vascular lumens in coronary segments next to obstructive coronary lesions, and more frequent completely occluded segments [81]. Diabetic patients not only present with an increased atherosclerotic burden but also an increased amount of fresh, lipid-rich coronary plaques, which have been proven to be more prone to rupture [82].

In addition, the coronary circulation of diabetic patients demonstrates a reduced ability to adapt to significantly obstructive lesions. Diabetic patients show a reduction in coronary collateral development, which provides intrinsic pathways from one coronary segment or artery, past an obstruction [83]. Coronary artery remodeling is a frequently encountered early compensatory enlargement at atherosclerotic sites for maintaining proper flow. However, newer imaging modalities like intravascular ultrasound have demonstrated that the coronary arteries in the diabetic population are less likely to undergo the proper remodeling pathways in response to atherosclerosis [84].

Reduced arterial healing is an important thrombogenic factor in diabetic patients who undergo PCI, which can produce an increase in stent strut exposure, an important pathophysiological mechanism correlated with ST [85]. In addition to this, improper stent size selection and malposition are more frequent, considering the diffusely affected and calcified coronary arteries of these patients. In the case of diffuse coronary artery disease, underestimation of true vessel diameters is a common challenge, as segments considered a “reference” may also be affected. As discussed above, there are several pathophysiological pathways that play a role in increasing the risk of ST in diabetic patients. All those mechanisms play an important role not only in the incidence of ST in this special population, but also in its timing.

There have been several trials comparing the outcomes of percutaneous coronary intervention in diabetic and nondiabetic populations. In the BIOSCIENCE trial, which included 2119 patients (22.9% of them with diabetes), there was no statistically significant difference in EST between diabetic and nondiabetic patients (1.9 versus 2%, HR = 0.91, *p* = 0.81) [86]. However, rates of LST were higher in the diabetic population (1.7% versus 0.9%, HR = 1.95, *p* = 0.13). It is also important to note that, in this study, the biodegradable polymer sirolimus-eluting stent (SES) showed a similar rate of ST to the newer generation non-erodible polymer–based everolimus-eluting stent (EES) in this high-risk population.

Additionally, a study that included 12,347 consecutive patients (1575 with and 10,772 without diabetes) from the Western Denmark Heart Registry [87] showed no difference in overall EST between the two populations studied (0.8% versus 0.9%, RR = 1.02). On the other hand, a statistically significant difference was noted in overall LST (3% in diabetic patients versus 1.1% in nondiabetics, RR = 2.56). Indeed, late ST appears to exhibit a higher incidence in diabetic patients, as shown in the results from the E-FIVE registry [88], which included 8314 patients, 32.7% of which had diabetes mellitus. Late ST was observed in 0.9% of nondiabetic patients compared to 1.7% of diabetics (*p* = 0.007), while any differences in early ST did not reach statistical significance.

These findings were confirmed in a recent meta-analysis [89], which included 18,910 patients and compared EST and LST in patients with diabetes mellitus (5123 patients) and in patients without diabetes mellitus (13,787 patients). In both groups, a similar rate of EST was observed (with an OR of 1.30); however, the incidence of LST was significantly higher in patients with diabetes mellitus, with an OR of 1.95. These differences in timing may result from the unique pathophysiology of ST in this population of patients, as well as the various comorbidities frequently associated with diabetes mellitus [90].

Nevertheless, a recent analysis of the Victorian circadian cardiac outcome registry, which included 43,209 patients (out of which 22.5% had DM), appeared to show statistically significant higher rates of early ST in this population, with DM being an independent predictor of early ST [91]. It is important to note, however, that the timing of ST in the diabetic population remains controversial, with investigations into the role of EST being insufficient.

Since the introduction of DES into clinical practice, increasing interest has been shown in comparing the safety and efficacy of different generations of DESs in diabetic patients with coronary artery disease. Several randomized trials and meta-analyses have since attempted to review the efficacy of these different DESs in the diabetic population. A recent meta-analysis, which included 4047 diabetic patients, compared the safety and efficacy of everolimus-eluting stents (EES), sirolimus-eluting stents (SES), and paclitaxel-eluting stents (PES) in this population. EES induced a lower rate of ST than SES, or PES, or the pooled SES and PES group (RR = 0.39, *p* = 0.0006) [92].

The risk of ST has also been evaluated in patients with prediabetes. A recent post hoc analysis of the BIO-RESORT and BIONYX clinical trials found comparable rates of ST in patients with prediabetes and normoglycemia, but higher rates occurred in the diabetic population (1.6% in diabetics compared to 1.1% in prediabetes and 0.4% in normoglycemic patients) [93].

Increased rates of ST, restenosis, and cardiovascular morbidity and mortality in patients with DM present a unique opportunity for new medical therapies to improve outcomes in this high-risk population.

Colchicine, a particularly potent anti-inflammatory agent, has recently emerged as a promising novel therapeutic agent to reduce adverse cardiovascular events in patients undergoing PCI. Colchicine’s anti-inflammatory effect is thought to be caused by inhibition of the NLRP3 inflammasome, which has been shown to play an important role in plaque development, progression, and destabilization [94]. In the colchicine cardiovascular outcomes trial (COLCOT) of 4745 patients, those receiving 0.5 mg of colchicine daily showed a statistically significant decrease in the composite of ischemic cardiovascular events (death from CV cause, resuscitated cardiac arrest, MI, stroke, and urgent revascularization), as well as a reduction in urgent hospitalization for angina leading to coronary revascularization [95]. It is of note, however, that this trial did not report rates of ST nor perform subgroup analyses of diabetic patients compared to non-diabetics. Thus, the role of colchicine in this population remains uncertain.

Coronary artery disease is virtually universally encountered in patients with diabetes mellitus, compared to nondiabetics, and usually carries a worse prognosis. The unique pathophysiological mechanisms of atherosclerosis in patients with diabetes produce a foundation for understanding their response to both medical therapy and revascularization. The incidence of MACE, cardiovascular mortality, and post-PCI ischemic events, including ST, appears to be higher in this population, and novel therapeutic agents, as well as advancements in stent technology, can be of particular importance in alleviating this problem. With this background in mind, clinical outcomes can potentially be improved in this high-risk group.

## 9. Clopidogrel Unresponsiveness

Coronary ST following PCI can be triggered by clopidogrel unresponsiveness, which has an estimated prevalence ranging from 5 to 44% [96]. The extent of clopidogrel unresponsiveness appears to be even higher with the use of clopidogrel generic bisulfate than that of the original formula [97]; this complication of antiplatelet therapy is highly prevalent also among the population with ischemic stroke/transient ischemic attack, where it is associated with poorer outcomes [98].

According to a study performed on 115 Iraqi subjects, in which 18.3% of the patients were non-responders to clopidogrel therapy, diabetes mellitus, hypertension, obesity, and male sex favor the occurrence of this phenomenon [96].

The association of aspirin and clopidogrel non-responsiveness with gender has also been confirmed by the results of the study that Pandey CP et al. performed on 207 patients with myocardial infarction who were receiving DAPT. In addition, in this study, genetic polymorphisms in thromboxane B2, glycoprotein VI (GPVI T>C) and the cytochrome P450-CYP2 family (CYP2C19*2 G>A) were significantly associated with non-responsiveness to antiplatelet therapy [99].

Clopidogrel it is catalyzed into its active metabolite by Cytochrome 450 (CYP2C19, CYP2B6, CYP1A2, CYP3A4, and CYP2C9) and Paraoxanse-1 (PON-1), and it appears that CYP2C19 plays the most important role in the therapeutic response to clopidogrel. The presence of the CYP2C19*2 mutant allele was associated with an increased risk of major adverse cardiovascular events (MACE) and decreased responsiveness to clopidogrel among 160 Chinese subjects with ischemic heart disease and PCI. Polymorphism in PON-1, although associated with a decreased platelet response, does not seem to result in an increase in the number of MACE [100]. The CYP2C19*3 mutant allele also proved to have a significant association with clopidogrel unresponsiveness [101]. CYP2C19 loss-of-function alleles are more prevalent in women, as emphasized by the results of the TAILOR-PCI (Tailored Antiplatelet Initiation to Lessen Outcomes Due to Decreased Clopidogrel Response After Percutaneous Coronary Intervention) trial [102].

Nevertheless, these genetic observations count for only a small percentage of the noticed clopidogrel resistance, and further evidence of the pharmacogenomics involved in the absorption and metabolism of clopidogrel is needed [103].

It has been suggested that modifications in platelet function also influence the responsiveness to clopidogrel, and that a high percentage of immature platelet fraction predicts a low response to clopidogrel [104]. High residual platelet reactivity also appears to modulate clopidogrel unresponsiveness, and Cirillo P et al. have suggested that the use of colchicine can inhibit platelet aggregation in non-responders. However, the results of this in vitro analysis need further validation with in vivo studies [105]. The measurement of platelet P-selectin also appears be useful in evaluating the efficacy of antiplatelet therapy and the rate of clopidogrel non-responders [106].

Fortunately, in non-responders to clopidogrel therapy, including those with the presence of CYP2C19*2 or CYP2C19*3 mutant alleles, ticagrelor and ticlopidine seem to be useful alternatives [107]. Ticagrelor appears to be more efficient than high-dose clopidogrel [108].

## 10. Genetics

As mentioned above, CYP2C19*2 loss-of-function alleles, as well as CYP2C19*3 mutant alleles, increase the risk of coronary ST following percutaneous coronary intervention in patients treated with clopidogrel, as these subjects are not able to activate clopidogrel. In Saudi Arabia, it has been reported that 20% of the population carries the CYP2C19*2 mutant allele. To improve the cardiovascular outcomes of these patients, it appears that bedside testing of the CYP2C19 gene can be useful [109].

In China, a pharmacogenetic-driven algorithm has been developed and tested in 1757 patients, with the purpose of guiding DAPT, and it appears that it can mitigate the risk of ST if implemented in clinical practice [110].

The utility of implementing a genotype-guided P2Y12 inhibitor therapy in clinical practice has been also emphasized by a Bayesian analysis of TAILOR-PCI (Tailored Antiplatelet Initiation to Lessen Outcomes Due to Decreased Clopidogrel Response After Percutaneous Coronary Intervention) trial [111].The post hoc analysis of the TAILOR-PCI resulted in the development of another risk score, namely the ABCD-GENE score (Age, Body Mass Index, Chronic Kidney Disease, Diabetes, and Genotyping), which also proved to be useful for the identification of patients at high risk of ST [112].

The implementation of a CYP2C19 genotype-guided therapy does not seem to be inferior to antiplatelet treatment with ticagrelor or prasugrel, as far as the risk of thrombotic events is concerned, and it appears to even decrease the risk of bleeding [113].

Apart from CYP2C19*2 or CYP2C19*3 mutant alleles, polymorphism in the P2Y12 purinoreceptor appears to also be associated with an increase in the risk of ST following PCI, as emphasized by the results of a meta-analysis on 10 studies and 10.810 clopidogrel-treated subjects [114]. Flavin-containing monooxygenase 3 (FMO3) rs1736557 polymorphism has a complementary influence on the response to clopidogrel therapy, and it appears that the FMO3 rs1736557 AA genotype increases the potency of clopidogrel [115].

Nevertheless, not only unresponsiveness to clopidogrel, but also to aspirin, can result in coronary ST following PCI. Regarding the genetics involved in the therapeutic response to aspirin, it has been suggested that homozygous GUCY1A3 (rs7692387)-risk allele carriers present a high risk of coronary ST within 1 month of stent implantation, due to an increase in on-aspirin platelet reactivity [116].

Understanding drug metabolism-related genetic polymorphism in patients treated with DAPT following PCI can therefore result in an improvement in the cardiovascular outcomes of this high-risk population.

The most important patient-related factors that may definitely increase the risk of ST are listed in Table 2.

## 11. Conclusions

ST, especially when it occurs suddenly, can have life-threatening consequences if not promptly addressed. In such cases, an immediate referral to a cardiologist is necessary for further examination and appropriate treatment. The optimal duration of DAPT after PCI remains uncertain, and may vary across different patient populations. Nevertheless, the occurrence of early ST has been shown to be associated with higher rates of death inside the hospital and within 30 days in patients who have primary PCI. Although PCI has seen notable advancements in terms of safety and efficiency, the occurrence of ST persists when using second-generation DESs. The appearance of early ST is associated with higher rates of in-hospital and 30-day death among patients undergoing primary PCI. Lower LVEF is correlated with both EST and LST. The occurrence of ST may also be influenced by genetic factors. Considering that these patients face an elevated risk of periprocedural myocardial infarction and ST following drug-eluting stent implantation, it is necessary to have a comprehensive CAD management. This includes state-of-the-art percutaneous coronary intervention, an appropriate antiplatelet therapy regimen tailored to the patient’s ischemic and bleeding risk, lifestyle intervention, intensive medical therapy, and thorough patient education. These measures should be considered essential for clinicians aiming to optimize the care and outcomes of CAD patients. In conclusion, determining the primary component most strongly associated with ST is challenging. However, particular emphasis should be placed on individuals exhibiting male gender, hypertension, diabetes, poorer pre-PCI LVEF, and a history of smoking. A comprehensive strategy for coronary revascularization has the potential to mitigate complications and optimize long-term outcomes in the management of ST.

## Figures and Tables

**Figure 1 jcm-12-07367-f001:**
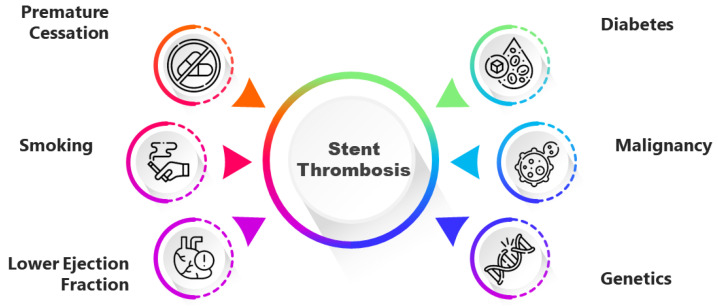
Patient-related factors that may increase the risk of stent thrombosis.

**Table 1 jcm-12-07367-t001:** Classification of stent thrombosis based on stent type, event timing and event certainty.

Stent Thrombosis Classification	Criteria		
**Stent type**	Bare metal stents	First-generation drug-eluting stents (DES)	Second-generation drug-eluting stents (DES)
**Event timing**	Early stent thrombosis (within 24 h)	Late stent thrombosis (1 to 12 months)	Very late stent thrombosis (after 12 months)
**Event certainty**	Definite: ACS with angiographic or autopsy confirmation of ST	Probable: Unexplained death within 30 days of stent implantation without autopsy AMI in the territory of a target vessel where stent was implanted but without angiographic confirmation	Possible: Unexplained death after 30 days of stent implantation without autopsy

**Table 2 jcm-12-07367-t002:** Patient-related factors that may increase the risk of stent thrombosis.

**1.** **Premature cessation of dual antiplatelet therapy** Shorter DAPT durations after ACS were associated with higher rates of ST and myocardial infarction [8,9]; No differences between short-duration and standard-duration DAPT in terms of effectiveness results, with short-term DAPT being an option for patients at high risk of bleeding [3]; A single month of DAPT was non-inferior to the long-term DAPT in terms of the risk of MACE [4,11,13]; DAPT withdrawal was not linked to poorer 1-year TLF following COMBO stenting [17]; Younger patients had a higher cardiovascular risk associated with disruption, whereas those over 75 did not [18]; The effect of stopping DAPT after 9 months following PCI is limited to cardiac events, and is generally safe [19]; the most significant factor associated with ST was the early discontinuation of clopidogrel [20].
**2.** **Percutaneous coronary intervention for acute coronary syndrome** Male gender, hypertension, diabetes, lower pre-PCI LVEF, Killip Class, and pre-procedure TIMI flow 0 are indicators of individuals susceptible to ST [23]; The risk of ST was found to be significantly higher in people with STEMI, compared to people with stable coronary artery disease [6,25,26,27,28]; Insufficient platelet inhibition is an important factor risk for ST [6,29]; In lowering ST, second-generation DESs exceeded first-generation DESs [30]; At five years, rates of MACE (target lesion revascularization, TVR, recurrent MI, and ST) were significantly lower in the EES group, compared to the SES group [31]; STRS is an independent predictor of ST after primary PCI [33]; The incidence of early ST increases in-hospital and 30-day mortality in patients receiving primary PCI [34].
**3.** **Smoking** Smoking is associated with a higher risk of MACE in females [37,38]; In patients who are current smokers, there were no differences between the STEMI and NSTEMI groups at 1 month or 2 years after index PCI [39]. Smoking was an independent predictor of MACE in patients undergoing PCI [36,41,42,43,46]; We discussed the “smoker’s paradox”, whereby the idea of smokers experiencing lower rates of adverse events following MI and after PCI is not supported [36,41,42]; Smokers with significant on-treatment platelet reactivity had a greater likelihood of ST [36,41].
**4.** **Lower ejection fraction** ST rates increase significantly as LV function declines, especially with at least a moderate systolic dysfunction [47,49,50]; Lower LVEF is correlated with both EST and LST [51]; HPR serves as a risk factor for subsequent cardiovascular events in patients undergoing treatment with aspirin and/or clopidogrel, generating a reduced effectiveness of the antiplatelet therapy [53].
**5.** **Malignancy** Cytostatic and cytotoxic chemotherapeutic agents may hinder the process of stent endothelialization, secondary to a diminished capacity for forming endothelial progenitor colonies, particularly during the acute phase of treatment [66,67,68]; Patients undergoing thoracic EBRT have an increased risk of developing ischemic heart disease, and the risk is closely correlated with the average radiation dose absorbed by the heart [69,70]; Malignancy is an independent risk factor for both EST and LST, high-thrombotic-risk cancers being associated with a greater risk of ST in the first year after PCI [51,73,74]; The rate of ST is nearly three times higher in patients with cancer [74].
**6.** **Diabetes mellitus** Increased platelet aggregation, a pro-thrombotic state due to platelet hyperactivity, endothelial dysfunction, and the anatomical features of CAD are risk factors of ischemic events in diabetes patients [76,81,83,84]; Platelet hyperreactivity, a worse response to P2Y12 inhibition, and less downregulated platelet reactivity in diabetic patients play significant roles in ST [78,79]; Reduced arterial healing, improper stent-size selection, and inadequate stent apposition are all risk factors associated with ST [85,86].
**7.** **Clopidogrel unresponsiveness** Genetic polymorphisms in thromboxane B2, glycoprotein VI and the cytochrome P450-CYP2 family were significantly associated with non-responsiveness to antiplatelet therapy [99]; Mutant alleles of CYP2C19*2 and CYP2C19*3 were linked to reduced responsiveness to clopidogrel [100,101].
**8.** **Genetics** The ABCD-GENE score proved to be useful for the identification of patients at high risk of ST [112]; Polymorphism in the P2Y12 purinoreceptor is associated with an increase in the risk of ST following PCI [114].

DAPT—dual antiplatelet treatment; TLF—target lesion failure; ST—stent thrombosis; PCI—percutaneous coronary intervention; LVEF—left ventricle ejection fraction; TIMI—thrombolysis in myocardial infarction; STEMI—ST-segment elevation myocardial infarction; DES—drug-eluting stent; MI—myocardial infarction; TVR—target vessel revascularization; EES—everolimus-eluting stent; SES—sirolimus-eluting stent; PES—paclitaxel eluting stents; NSTEMI—non-ST segment elevation myocardial infarction; HPR—high platelet reactivity; EBRT—external beam radiation therapy; CAD—coronary artery disease; ABCD-GENE score—Age, Body Mass Index, Chronic Kidney Disease, Diabetes, and Genotyping; STRS—Stent Thrombosis Risk Score; EST—early stent thrombosis; LST—late stent thrombosis.

## Data Availability

Data are contained within the article.

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
