# Peer review of "Patient-Related Factors Predicting Stent Thrombosis in Percutaneous Coronary Interventions"

_jcm, 2023, doi:10.3390/jcm12237367_

Round 1
Reviewer 1 Report
Comments and Suggestions for Authors
This review article by Romanian authors gave us a wide view of possible causes of stent thrombosis. They described in detail every risk factor which could cause life-threatening complication, like stent thrombosis.
However, I have some comments.
1. In the section Premature cessation of dual antiplatelet therapy, line 105, the sentence “Additionally, the incidence of ST decreased 105 from 35.4% to 11.7% when clopidogrel was stopped during the first 180 days following 106 index-PCI from 35.4% to 11.7% when it was stopped within the first 30 days” does not have too much sense because the numbers 35.4% to 11.7% were mentioned twice.
2. In the section Premature cessation of dual antiplatelet therapy, the paragraph from lines 128-156 should be shorter, authors should just mention main findings of the MASTER DAPT, GLOBAL LEADERS and STOPDAPT2 trails.
3. In the section Percutaneous coronary intervention for acute coronary syndrome paragraph from lines 224 to 246 and paragraph from lines 254 to 279 should also be shorter.
4. In the section Smoking, the first paragraph should also be shorter and focused on the facts that smokers had higher rate of recurrent MI and death after PCI and that there were no significant difference of outcomes among smokers who had STEMI and NSTEMI.
Author Response
Dear Reviewer,
Our team would like to sincerely thank you for the positive manuscript appreciation and for all your valuable suggestions. We have applied all the suggested text modifications, and we firmly believe your input was crucial to the global value of the manuscript. We have carefully addressed your comments and have revised our manuscript accordingly. Revised portions are highlighted in red.
Please see the attachment.

Reviewer 2 Report
Comments and Suggestions for Authors
Authors submitted a review aimed to consider each procedural and patient-related factors associated with stent thrombosis.
The article is original since many risk factors were considered, and each factor was deeply reviewed.
Nevertheless, the method used to select risk factors considered should be reported. Why stent malapposition was not included?
A section “methods” should be added to show how the review of the literature was performed. (i.e. search keywords)
Conclusions should be modified in order to make the reader understand which of the patient- and procedures- factors considered are more consistent with the risk of stent thrombosis.
In the present work the authors aimed at giving a focus on several patient-related-factors and their correlation with the risk of stent thrombosis. The current paper is very interesting but the English language and style need to be revised.
Comments on the Quality of English Language
Please find some suggestion about the corrections needed.
At line 1: insert the acronym PCI;
At line 50: pay attention to rimove the dash in "des-ending";
try to reformulate the sentence at lines 51-53, maybe using brackets it could be more understandable.
At line 73: "Percuteneous coronary intervention" was already written at line 1, insert directly PCI;
Line 82: insert the reference of guidelines of the ESC;
Line 163: use the past;
Line 207: use the past;
Line 255: insert the reference of Guidelines;
Line 268: maybe "showed" is more appropriate;
Line 385: you can write directly LEVF;
In line 41 you already wrote the acronym ACS for acute coronary syndrome. Many times in the text (lines 57, 82, 103 and so on...) you used the long formula: remove it and use "ACS";
Line 467, 473, 480: remove the dash in re-gimens, fac-tors, dis-ease.
Finally, the manuscript has to be improoved and a complete revision of the English is required.
Author Response

(The authors gave the same response as above.)

Round 2
Reviewer 2 Report
Comments and Suggestions for Authors
Authors improved the manuscript following the suggestions proposed obtaining a good result. The section about Methods was absolutely necessary and also Conclusions now are consisting with the text.
Comments on the Quality of English Language
Even though the readability of the text was improved, a revision of the English is still needed. Please pay attention to remove the dash that are inappropriate and check all the acronyms (i.e. Stent thrombosis at line 36, you should introduce the acronym and remove the extended version from line 53).
Author Response
Dear reviewer,
We very much appreciate this opportunity to submit our revised manuscript. We have carefully addressed your comments, and have revised our manuscript accordingly. On behalf of all authors of this work, we want to thank you for your recommendations and for your appreciation.
We greatly appreciate the comments and suggestions and hope that our manuscript will be acceptable for publication with these alterations. We remain prepared to revise our manuscript further should it be required.
Kind regards,
All authors